# Rheological modification of partially oxidised cellulose nanofibril gels with inorganic clays

**Saffron J. Bryant** [1,2]*, **Vincenzo Calabrese**[1], **Marcelo A. da Silva**[1], **Kazi M. Zakir Hossain**[1], **Janet L. Scott**[1,3], **Karen J. Edler**[1]*

**1** Department of Chemistry, University of Bath, Claverton Down, Bath, United Kingdom, **2** School of Science, RMIT University, Melbourne, Victoria, Australia, **3** Centre for Sustainable Chemical Technologies, University of Bath, Claverton Down, Bath, United Kingdom

* saffron.bryant@rmit.edu.au (SJB); k.edler@bath.ac.uk (KJE)

**Data Availability Statement:** Data supporting this work are freely accessible in the Bath research data archive system at DOI: 10.15125/BATH-00791.

## Abstract

This study aimed to quantify the influence of clays and partially oxidised cellulose nanofibrils (OCNF) on gelation as well as characterise their physical and chemical interactions. Mixtures of Laponite and montmorillonite clays with OCNF form shear-thinning gels that are more viscous across the entire shear range than OCNF on its own. Viscosity and other rheological properties can be fine-tuned using different types of clay at different concentrations (0.5–2 wt%). Laponite particles are an order of magnitude smaller than those of montmorillonite (radii of 150 Å compared to 2000 Å) and are therefore able to facilitate networking of the cellulose fibrils, resulting in stronger effects on rheological properties including greater viscosity. This work presents a mechanism for modifying rheological properties using renewable and environmentally-friendly nanocellulose and clays which could be used in a variety of industrial products including home and personal care formulations.

## Introduction

Environmental concerns have led to the search for biologically-friendly and renewable replacements for common household products. For example, many personal care and household cleaning products use rheological modifiers such as polymer additives or surfactants derived from oil sources, which are either environmentally damaging or non-renewable [1]. In contrast, cellulose is non-toxic, renewable, and abundant, and has been shown to act as a rheological modifier in aqueous formulations [1–3].

Similarly, clays offer an environmentally friendly method of rheological modification [4–6]. Previous research examined the combination of clays and cellulose to produce films [7–9], papers [10–12], hydrogels [13], and nanocomposites [14–16]. In the case of films, addition of clay led to increased tensile strength [7], and oxygen barrier properties [9]. Addition of sodium montmorillonite to nanopapers improved barrier properties and increased flame resistance [11]. For dispersions, it has been shown that mixing cellulose nanocrystals and microfibrilated cellulose with montmorillonite conveys shear-thinning behaviour and also increases viscosity through formation of a network [17]. All of this research demonstrates that clay platelets can interact with cellulose fibrils to create networks. However previous work has mostly focused on papermaking or films. One recent paper explored the rheological properties of mixtures of

**Funding:** KJE (principal investigator) and JLS (co-investigator) received an award from the Engineering and Physical Sciences Research Council to fund this project, which employed SB, MADS and KMZH (grant no. EP/N033310/1) https://epsrc.ukri.org/. VC received funding from the University of Bath to support his PhD candidature. The funders had no role in study design, data collection and analysis, decision to publish, or preparation of the manuscript.

**Competing interests:** The authors have declared that no competing interests exist.

Laponite and oxidised cellulose nanofibrils and found an additive effect [18]. However, to date the effect of different types of clays in combination with oxidised cellulose nanofibrils to make gels has not been investigated and so further characterisation of these dispersions is required to better understand the underlying interactions.

Rheological modification is important in household and personal products for both aesthetic reasons e.g. silkiness and overall appearance, but also for technical reasons such as processing, ease of application and preventing sedimentation during storage [19]. There are different aspects of rheology, but the primary parameter is viscosity. In many industrial products, the change of a formula's viscosity with changing shear is a key characteristic. Often it is desirable for the formula to have a low viscosity during shear (e.g. during application or pumping) but a high viscosity at rest (to prevent sedimentation and spillage) i.e. to have shear-thinning properties [19].

This work utilised TEMPO-mediated oxidised cellulose nanofibrils (OCNF) which have a large aspect ratio (5–10 nm cross-section and 100 nm to several μm in length) and negatively charged carboxyl groups on the fibril surface, making them easy to disperse due to a high surface charge (-60 mV in ζ potential) [20,21]. OCNF was chosen over cellulose nanocrystals because the functionalisation method of OCNF is more environmentally benign and more atom efficient than production of CNCs [20]. Furthermore, OCNF has been shown to be biodegradable [22]. OCNF was combined with two different clays, namely Laponite and montmorillonite, to explore interactions and to understand the effect of those interactions on rheological behaviour. Laponite is a synthetic clay while montmorillonite is a natural clay. Both clays have layers composed of three sheets; two outer tetrahedral silica sheets and a central octahedral sheet. In water, the platelets generally have negatively charged faces, and positively charged edges. Both Laponite and montmorillonite platelets are 1–4 nm thick, but Laponite platelets are 30 nm in diameter while montmorillonite platelets can have diameters up to 500 nm [5,23]. These clays have low toxicity, even with oral ingestion, and so are ideal candidates for biocompatible product development [24,25].

The charged nature of the clay particles provides two potential avenues for interaction with the OCNF fibrils. Specifically, the positively charged edges of the particles will have attractive interactions with not just the negatively charged faces of other particles, but also the negatively charged OCNF fibrils, thus building an inter-connected network. Furthermore, the negatively charged faces of the particles may contribute to exclusion/repulsive interactions with the OCNF fibrils which will also contribute to the rheological properties of the gels. A previous study examining mixtures of Laponite and OCNF highlighted the importance of clay concentration to the relative importance of clay-clay vs. clay-fibril interactions for rheological properties [18].

This research aimed to quantify the effects of OCNF and either natural or synthetic clays on the rheological properties of gels. The research presented here demonstrates that shear-thinning gels with a range of rheological properties can be obtained by varying either the OCNF concentration from 0.75 to 2 wt%, the clay concentration from 0.5 to 2 wt%, or by changing from a synthetic to a natural clay. These parameters open up a region of formulation space with viscosity profiles relevant to a range of aqueous products from paints to cosmetics.

## Method

Oxidised cellulose nanofibrils were prepared via TEMPO-mediated oxidation as previously described [3,20,21]. According to previous work, the degree of oxidation in terms of the carboxylate group content of the nanofibrils as a function of the number of anhydroglucose units was 25%.[26] (See S1 File, page 1 for more details).

OCNF was purified to remove residual salts and preservatives by dialysis against deionised water (18.2 MΩ·cm) as previously described [3]. Following dialysis, the OCNF was freeze-dried and then resuspended in deionised water to 2 wt%. OCNF was dispersed with sonication (Ultrasonic Processor, FB-505, Fisher– 550 W), at 30% amplitude with 1 s on 1 s off pulses, for ~1 hr or until the dispersion became transparent (pH = 7). This allowed redispersion of the fibrils without sedimentation.

Laponite and montmorillonite were provided by Rockwood Additives Limited (now BYK Additives Ltd) and used as received. Rheological measurements of each sample were made using a stress-controlled Discovery Hybrid Rheometer, Model HR-3 (TA Instruments) with a sand-blasted 40 mm parallel plate geometry and a 1 mm gap. The temperature was kept at 25˚C using a Peltier unit (±0.1˚C). A thin layer of mineral oil was applied to the edge of the sample to prevent evaporation. Measurements were made as follows: frequency sweeps were performed at 0.1% strain covering an angular frequency from 0.01 to 50 rad/s (S1-S8 Figs in the S1 File), oscillation amplitude sweeps were performed at a fixed angular frequency of 6.28 rad/s across an oscillation strain range from 0.01 to 1000% (S9-S16 Figs in the S1 File), flow sweeps were performed at shear rates from 0.01 to 100 s$^{-1}$ (S17-S21 Figs in the S1 File). Frequency sweeps recorded the storage and loss moduli across a range of angular frequencies, amplitude sweeps gave an indication of the yield strain and stress, and flow sweeps demonstrated shear thinning behaviour. Measurements were performed in duplicate and the error bars presented in this work are based on the standard deviation between measurements. In many cases, the error is smaller than the symbol plotted on the graph.

Laponite or montmorillonite powder was added to OCNF dispersed in water at either 0.75. 1.5 or 2 wt% to get clay concentrations of 0.5, 1, or 2 wt% and mixed thoroughly. Preliminary tests demonstrated that gel formation is not instantaneous (S23 Fig in the S1 File), therefore, all rheological measurements reported here were performed 24 hrs after sample preparation to allow time for gelation and to ensure consistency across measurements.

SAXS measurements were performed on all combinations of OCNF/clays using an Anton-Parr SAXSpoint 2.0 provided by the Material and Chemical Characterisation Facility (MC$^2$) [27] equipped with a copper source (Cu K-α, λ = 1.542 Å) and a 2D EIGER R series Hybrid Photon Counting (HPC) detector. The sample detector distance was 556.9 mm covering q range of about 0.008–0.4 Å$^{-1}$. Samples were loaded into 1 mm quartz capillaries. Data was collected in four frames, with 300 s exposure per frame, then averaged and processed. Fitting was performed using SASView (Version 4.2.1, see http://www.sasview.org/ for more information) (models detailed in SI). The temperature was kept at 25˚C using a Peltier unit (±0.1˚C).

Experiments were designed to test the effects of OCNF concentration, clay concentration and clay type by varying each factor in turn while the others were kept constant.

## Results and discussion

### Gelation

One definition of a gel, in rheological terms, is a material where G' and G" show no frequency dependence over the probed frequency range, i.e. infinite relaxation time, and G'>G" (that is, tan(δ)<1). Frequency independence of G' and G" is normally observed in chemical or permanent gels. For physical gels, due to the transient nature of forces maintaining the gel network, some relaxation will be observed and G' and G" will show a weak frequency dependence [28]. Fig 1 shows the frequency sweeps of 2 wt% OCNF with either 1 wt% Laponite or 1 wt% montmorillonite. From the graph it is clear that the montmorillonite suspension has a stronger frequency dependency than the Laponite suspension, as demonstrated by the increase in both the storage and loss moduli with increasing angular frequency. S1-S7 Figs in the S1 File show the

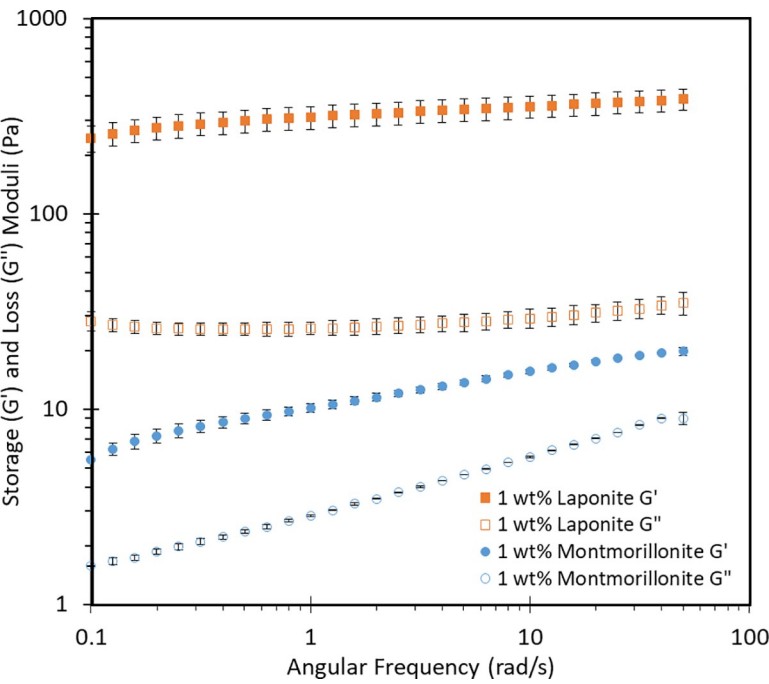

**Fig 1. Frequency sweep curve of 2 wt% OCNF samples.** OCNF (2 wt%) with either 1 wt% Laponite (orange squares) or 1 wt% montmorillonite (blue circles).

frequency sweeps of the clay-OCNF mixtures, demonstrating some frequency dependence, as expected for gels maintained by electrostatic repulsion, especially for mixtures containing montmorillonite.

The tan(δ) for a specific frequency and strain value offers a convenient way to follow changes in the solid/liquid character of the gels while ignoring frequency-dependent changes, therefore, we will focus mainly on changes in tan(δ). This parameter (tan(δ)) is defined as the ratio of the loss modulus to the storage modulus and gives a simple indication of whether the viscous or the elastic behaviour is dominating. For the purposes of this research, tan(δ) was taken from the frequency sweep curves (see Supporting information) at an angular frequency of 6.3 rad/s (equivalent to 1 Hz). A lower tan(δ) value (below 1) is associated with more solid or gel-like behaviour, while a higher value (tan(δ)>1) is associated with liquid-like behaviour. The tan(δ) and frequency dependence of OCNF on its own decreases with increasing OCNF concentration due to repulsive interactions between the fibres and excluded volume interactions [3]. The rheological properties of OCNF at 0.75 wt% could not be accurately measured because it is too liquid like (see S1, S9 or S21 Figs in the S1 File for examples of curves taken from liquid-like samples). At 1.5 wt% OCNF, the rheometer is operating near its lower limits because the dispersion is too liquid-like. However, the tan(δ) obtained was 1.25, while it was 0.55 for 2 wt% OCNF. Rheological measurements for OCNF at different concentrations are shown in the Supporting Information.

Combinations of Laponite with OCNF formed gels at all concentrations assayed, while for montmorillonite a minimal concentration threshold was observed for gel formation. Fig 2 shows the tan(δ) values for all combinations of Laponite and montmorillonite with OCNF. The tan(δ) decreased with increasing Laponite concentration at all OCNF concentrations tested, indicating that the formulations were becoming more solid-like. However, increasing the wt% of OCNF at a set Laponite concentration did not significantly change the tan(δ) value,

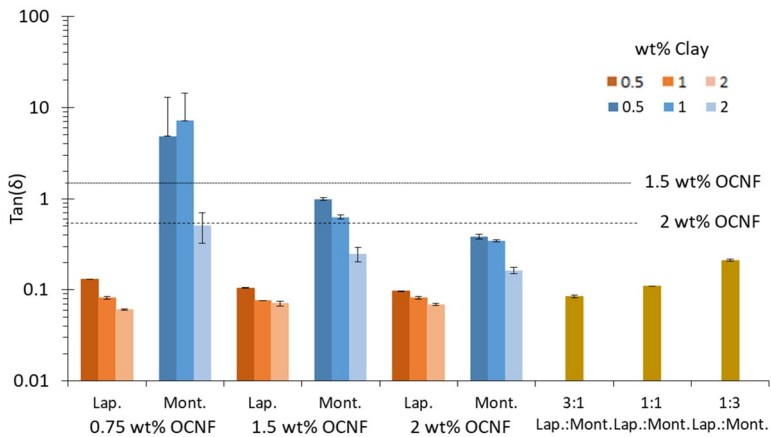

**Fig 2. The tan(δ) values of mixtures of Laponite and montmorillonite with OCNF.** The negative error bars of 0.5 and 1 wt% montmorillonite in 0.75 wt% OCNF go beyond zero and are therefore not displayed on a log plot.

showing that Laponite contributed more significantly to the solid-like behaviour and that further addition of OCNF had little effect. This may be because Laponite is a much smaller particle and therefore on a per-weight basis has a higher contribution to electrostatic interactions than OCNF fibrils. A similar effect was observed by Šebenik *et al*. (2020) and they attributed it to the fact that at low Laponite concentrations the clay platelets are acting as bridges between the OCNF fibrils, but at higher concentrations the interactions between clay platelets dominates over any contribution from the fibrils [18]. It should be noted that the rheology results presented in that paper showed greater gelation and viscosity than presented here, however that is most likely an aging effect as in that study, mixtures were stored for 20–80 days prior to measurement while here they were measured after just 24 hours. We show the evolution of rheological behaviour in S23 Fig in the S1 File and this evolution is expected to continue with increasing time.

According to previous work, in the presence of salt, the OCNF network has a mesh size of 30 ± 10 nm [3]. While the mixtures presented here do not contain salt, this previous value gives some idea of the mesh size that an OCNF network may form. The small Laponite particles (diameter 30 nm) would fit into this theoretical mesh size and form a bridge between fibres, consistent with other results presented in the literature [18]. In addition, the edges of Laponite platelets are positively charged. Therefore, attractive interactions between the edges of Laponite particles and the negatively charged groups on OCNF could create a network of fibrils with a more solid-like structure than is achieved by negatively charged OCNF fibrils on their own.

Montmorillonite was less effective at forming gels with OCNF than Laponite. At low OCNF and montmorillonite concentrations, very weak viscoelastic dispersions were formed with properties below the sensitivity of the rheometer. Therefore, the results for 0.5 and 1 wt% montmorillonite in 0.75 wt% OCNF have a large degree of inaccuracy. S1 and S2 Figs in the S1 File demonstrate the much greater frequency dependence and greater measurement error of mixtures with low montmorillonite concentrations, that is, a lack of gel formation. These mixtures visually behaved like liquids: flowing upon tilting of the vial, whereas mixtures of 2 wt% montmorillonite with all OCNF concentrations, and all montmorillonite concentrations in 2 wt% OCNF formed self-standing gels that did not flow upon tilting. Thus, the threshold for gel formation depends on both the montmorillonite and the OCNF concentration.

At 0.75 wt% OCNF, the error of montmorillonite mixtures was too high to draw conclusions from the tan(δ) values. At 1.5 wt% and 2 wt% OCNF, increasing montmorillonite concentration resulted in a decrease in tan(δ), i.e. the mixtures became more solid-like. Increasing the OCNF concentration at a fixed montmorillonite concentration also resulted in a decrease in tan(δ).

All of these results show that both OCNF and montmorillonite contribute to decreasing tan(δ) and increasing the solid-like behaviour of these mixtures. In all cases, at the same OCNF and clay concentrations, Laponite mixtures had lower tan(δ) values than montmorillonite mixtures. The measured ζ-potential of montmorillonite and laponite were very similar (see S1 File) and so does not account for the differences in behaviour. Montmorillonite platelets have the same charge arrangement (negative faces and positive edges) as Laponite but are much larger (300 nm diameter compared to 30 nm) and therefore would be less able to bring fibrils together. It is more likely that the large size of montmorillonite platelets would keep OCNF fibrils apart, rather than facilitating network formation. Previous rheological studies of different composite materials have also found that smaller filler size results in greater viscosity [29,30]. This is attributed to the greater number of particles for the same volume (or in this study, weight) and therefore greater surface area and more interactions between particles.

Mixtures of Laponite and montmorillonite were measured at 1.5 wt% OCNF to provide midpoint data. The total clay concentration was 1 wt%, with mixtures of either 3:1, 1:1, or 1:3 ratios of Laponite:montmorillonite. The 0.75 wt% Laponite sample had a tan(δ) almost indistinguishable from the sample made of 1 wt% Laponite in 1.5 wt% OCNF. The tan(δ) value of the 0.5 wt% Laponite + 0.5 wt% montmorillonite sample was the same as the tan(δ) of just 0.5 wt% Laponite with OCNF on its own, suggesting that montmorillonite is not influencing the tan(δ) of this sample at all. This is consistent with the results of OCNF+montmorillonite, where the tan(δ) even at the highest clay concentration is only marginally lower than that of OCNF on its own.

Decreasing the Laponite concentration with concurrent increase in the montmorillonite concentration resulted in an increase in the tan(δ) value. However, even at 0.75 wt% montmorillonite + 0.25 wt% Laponite, the tan(δ) value remained lower than that of the 1 wt% montmorillonite + OCNF, and in fact was more similar to 2 wt% montmorillonite in 1.5 wt% OCNF.

These results suggest that Laponite and montmorillonite do not have a synergistic effect on the rheological behaviour of OCNF mixtures. The tan(δ) values suggest that the behaviour may be additive, although it appears that Laponite has a greater influence on tan(δ) than montmorillonite, further confirming the results of the single clay samples.

## Shear thinning

All combinations of clays and OCNF exhibited the same shear-thinning behaviour over the range tested. Fig 3 shows the flow curves for 1.5 and 2 wt% OCNF on its own, and 1.5 wt% OCNF in combination with either Laponite or montmorillonite at 0.5 wt% (so total particle concentration is 2 wt% and therefore comparable to the 2 wt% OCNF sample). The flow curves for the other mixtures can be found in the SI.

As shown in Fig 3, OCNF on its own and in combination with Laponite or montmorillonite is shear thinning. The viscosity across the whole range is higher at 2 wt% OCNF compared to 1.5 wt% and when OCNF is combined with montmorillonite. Addition of 0.5 wt% montmorillonite caused only a marginal increase in viscosity, when compared to 1.5 wt% OCNF on its own. Laponite results in an increase in viscosity of an order of magnitude, compared to 2 wt% OCNF or 1.5 wt% OCNF with montmorillonite. This is consistent with the patterns observed for tan(δ) discussed above. Strong interactions between Laponite particles and OCNF leads to

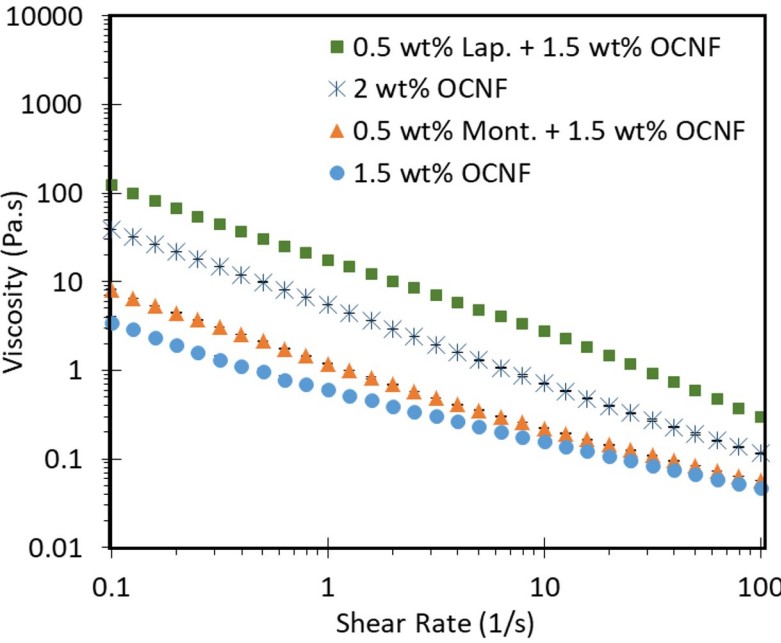

**Fig 3. Flow sweeps of OCNF and clays.** Flow sweep curves of 2 wt% OCNF (blue crosses), 1.5 wt% OCNF (blue circles), and 1.5 wt% OCNF with 0.5 wt% of either Laponite (green squares) or montmorillonite (orange triangles).

gelation, thickening and higher viscosities. These results are consistent with previous work which also showed an increase in viscosity upon combination of montmorillonite and cellulose fibrils [17,31].

Under shear flow, the gel structure is disrupted and the main contribution to viscosity comes from interactions between dispersed particles. As mentioned above, Laponite particles are much smaller than montmorillonite. Therefore, there are more Laponite particles than montmorillonite in the same weight percentage. This explains the higher viscosity of Laponite/OCNF systems shown in Fig 3, compared to montmorillonite/OCNF; the significantly greater number of particles in the Laponite mixture leads to more interactions and higher viscosity. The increase in viscosity on addition of montmorillonite is probably due solely to the presence of more particles as the flow sweep curve of 1.5 wt% OCNF + 0.5 wt% montmorillonite is only marginally higher than the curve for 1.5 wt% OCNF on its own. This further indicates that no strong interactions occur in montmorillonite mixtures.

Fitting models to the linear portion of flow curves, using the power law $\eta = K\dot{\gamma}^m$ (where h is viscosity, $\dot{\gamma}$ is the shear rate, K is the flow consistency index and m is the flow behaviour index) can give quantitative information about the systems for comparison [32] (see also description in Supporting information, page 15–17). S1 Table in the S1 File shows the best-fits for each combination of Laponite or montmorillonite with OCNF. The exponent (flow behaviour index) was similar for most mixtures of OCNF and Laponite, with values around 0.1, although it decreased with increasing Laponite concentrations, suggesting more prominent shear thinning behaviour. Mixtures with montmorillonite were much the same, also having values for the exponent around 0.1, except with 0.75 wt% OCNF, where the exponents were much closer to 1, suggesting much less shear thinning occurs in these low viscosity mixtures than in the other systems. The base number (flow consistency index) which is a measurement of the 'thickness' of a system followed the same patterns as those of viscosity and tan(δ) discussed above (see S1 Table in the S1 File), increasing with increasing clay or OCNF

concentrations and being much higher for systems with Laponite than montmorillonite, as expected from the other measurements.

These results demonstrate that the presence of the clay particles does not interfere with the inherent shear-thinning behaviour of OCNF, discussed elsewhere [1,33]. This may be useful for industrial applications as the clay particles provide an easy and cheap method of fine-tuning the viscosity of OCNF formulations, without destroying the important shear-thinning behaviour.

## Yield strain and stress

Amplitude sweeps were performed on all mixtures to explore the effect of Laponite and montmorillonite on the yield strain and stress of OCNF mixtures. Fig 4 shows the storage and loss moduli of Laponite and montmorillonite at 1 wt% in 1.5 wt% OCNF against oscillation strain. Amplitude sweeps for the other mixtures are given in the SI. The yield strain/stress is the strain (or stress) at which the gel network breaks down and the system starts flowing, here, we are taking the point at which the storage and loss moduli cross in the amplitude sweep as the yield strain (or yield stress, as shown in S12 Fig in the S1 File). Fig 5 shows the yield strain and stress of each mixture.

In Fig 4 there is an increase or overshoot of G" in the Laponite mixture. This was observed for all mixtures of OCNF with Laponite, and for some mixtures with montmorillonite (see Supporting information). This overshoot has been observed previously by our group for OCNF [34,35]. Strain thickening such as this is a result of resistance that comes about due to deformation of the system. For example, deformation of the OCNF network may lead to orientation which is combatted by repulsive electrostatic forces [35]. Similar behaviour has been observed in other systems, such as xanthan gum [36] and branched polymers [37].

For each OCNF concentration where measurements were possible, increasing the amount of clay resulted in a decrease in yield strain and an increase in yield stress. This suggests that

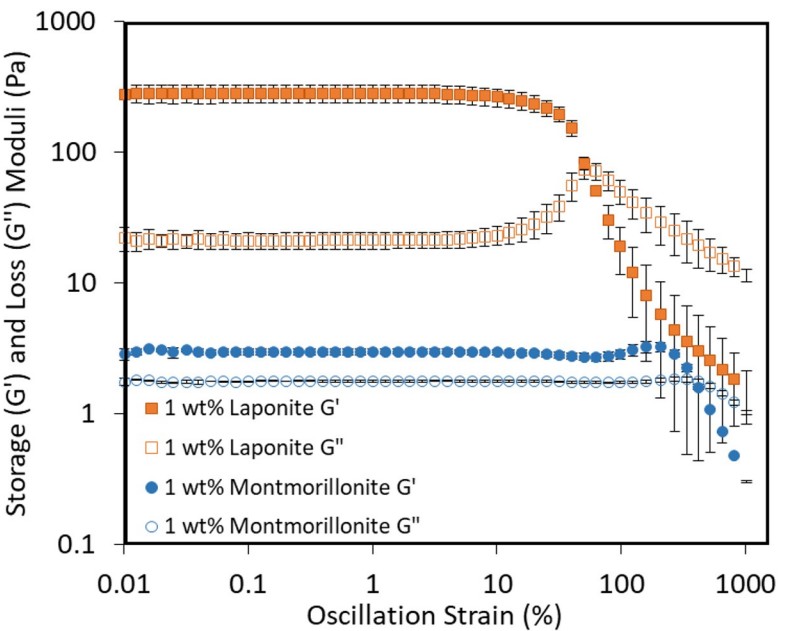

**Fig 4. Amplitude sweep curves of 1 wt% Laponite (orange squares) and 1 wt% montmorillonite (blue circles) in 1.5 wt% OCNF.** Storage moduli (G') shown with filled symbols, loss moduli (G") shown in open symbols.

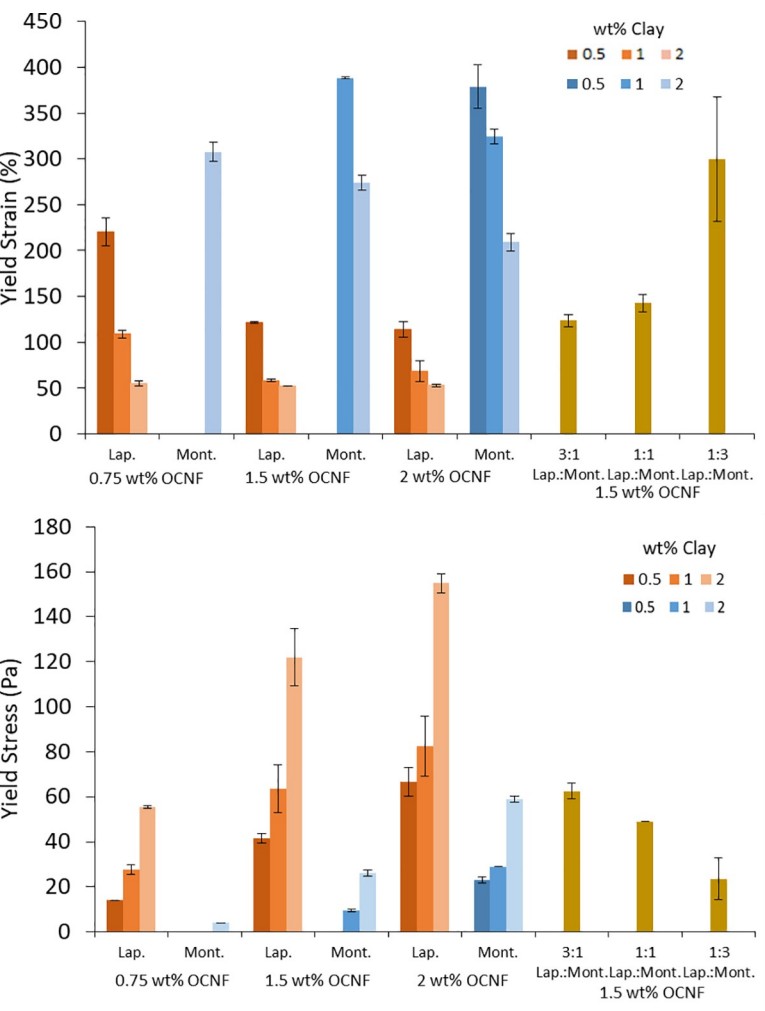

**Fig 5.** Yield strains (top) and yield stresses (bottom) of mixtures of Laponite and montmorillonite with OCNF, taken from the amplitude sweep curves (see Supporting information).

while mixtures with more clay became tougher (larger yield stress, and higher values of G' and G" as discussed above), they also became more brittle and harder to stretch (smaller yield strain). This may be because the platelets form bridges and connections between OCNF fibrils, thus reducing the ability of the network to stretch and move, forcing it to break at lower strains. This is consistent with previous work which showed an increase in yield stress with increasing Laponite concentrations in OCNF [18].

Similar to the tan(δ) values, the yield strain of mixtures with 2 wt% Laponite, did not change significantly upon increasing the OCNF concentration while the yield stress increased. It is possible that this is because at 2 wt% there are enough Laponite platelets to form bridges between all of the OCNF fibrils (even at 2 wt% OCNF), thus preventing stretching of the system, but addition of OCNF does allow the formation of more bridges, which increases the yield stress.

For 0.5 and 1 wt% montmorillonite in 0.75 wt% OCNF and 0.5 wt% montmorillonite in 1.5 wt% OCNF, the yield strain and stress could not be measured because G' and G" did not cross in the measured range. These samples were too liquid like to exhibit yield points with the current method. As the rheological properties of some of these mixtures were frequency

dependent, and therefore they were not gels, the yield strain and stress is meaningless for these systems. However for the other montmorillonite mixtures, similar to Laponite, increasing the concentration of montmorillonite caused a decrease in the yield strain and an increase in yield stress at a given OCNF concentration. Where comparisons are possible, it is evident that increasing the OCNF concentration at a fixed montmorillonite concentration also caused a decrease in the yield strain and increase in yield stress.

The yield strain of Laponite samples was lower than the equivalent montmorillonite sample while yield stress was higher. This may be because the smaller Laponite particles form better or more bridges between the OCNF fibrils, thus limiting the ability of the network to move and deform.

Mixtures of Laponite and montmorillonite had yield strains higher than Laponite on its own, but lower than montmorillonite on its own. Similarly, the yield stresses of mixtures were partway between that of the individual clays. This suggests an additive effect, without synergism, between the two clays. As with the tan($\delta$) values, the yield strains are closer to the values for Laponite than montmorillonite, suggesting a stronger influence of the smaller clay particles.

## Small angle X-ray scattering

Small angle X-ray scattering was used to assess the structure of dispersions of clay particles and OCNF. Scattering patterns for all mixtures analysed can be found in the Supporting information (S24-S33 Figs in the S1 File). Fig 6 shows the scattering patterns of Laponite and montmorillonite in water and in 1.5 wt% OCNF. It also shows the scattering pattern of 1.5 wt% OCNF on its own. Fitting parameters are given in S2 Table in the S1 File.

1.5 wt% OCNF SAXS data was fitted using an elliptical cylinder model, consistent with previous work [3]. The length parameter was set at 1000 Å (outside of the range accessible in this q range) and the radii allowed to vary. The minor radius of the fit was 12.4±2.0 Å and the axis ratio was 4.7±0.5. This is consistent with previous work from this group which found a minor radius of 14±1 Å and a major radius of 51±1 Å [3].

Both Laponite and montmorillonite were fitted with a cylindrical disk model with a length of approximately 10 Å, which is consistent with the literature values of 10–40 Å [5]. Laponite

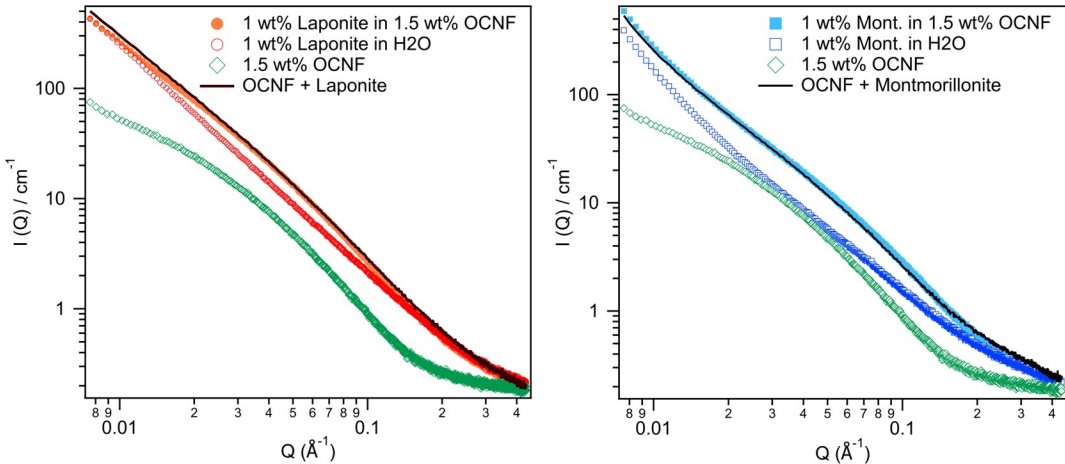

**Fig 6. SAXS scattering data from OCNF and clay dispersions.** A) 1.5 wt% OCNF, 1 wt% Laponite in water, and 1 wt% Laponite in 1.5 wt% OCNF. The black line is the scattering of 1.5 wt% OCNF and 1 wt% Laponite in water added together. B) 1.5 wt% OCNF, 1 wt% montmorillonite in water, and 1 wt% Laponite in 1.5 wt% OCNF. The black line is the scattering of 1.5 wt% OCNF and 1 wt% montmorillonite in water added together, with normalization for intensity.

could be fit with a radius of 250 Å or greater (outside of the probed q range) Å and montmorillonite with a radius of >1000 Å. In the case of Laponite this is larger than the literature value (150 Å), however, it is possible that platelets are somewhat aggregated e.g. two or more platelets together. The radius value for montmorillonite is arbitrary as this size is well outside of the probed q range and so could be varied without significantly altering the fit (e.g. down to 1000 Å or up to 3000 Å) [5].

As shown in Fig 6, the mixtures of clays and OCNF give scattering patterns that are additive of the individual components. Therefore, fits can be achieved by simple combination of the elliptical cylinder fit of OCNF, and the disk fit of the clay particle. This demonstrates that the change in rheological behaviour described above must occur due to interactions between OCNF and clay platelets arising at larger length scales than observed in the scattering patterns, most likely through interactions of OCNF and clay aggregates, rather than individual nanoparticles. This is illustrated in Fig 7.

Similarly, comparison of the other scattering patterns, e.g. of increasing clay concentration with a fixed OCNF concentration, or increasing the OCNF concentration at a fixed clay concentration reveal no significant changes in scattering (see Supporting information, pages 19–25) or interactions at the measured length scales. Scattering continues to meet expectations for additive scattering for all samples tested, including mixtures of Laponite and montmorillonite. These results further confirm that in these dilute systems, the interactions between OCNF and clay particles are happening at length scales that are not accessible by SAXS.

## Conclusion

The work presented here describes a way of modifying the rheological properties of hydrogels made from oxidised cellulose nanofibrils using inorganic clays. Addition of clay particles

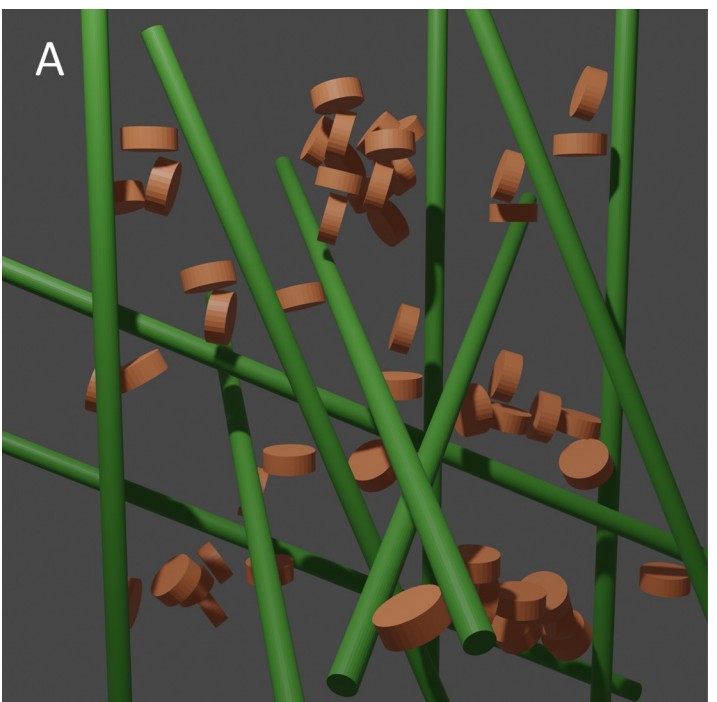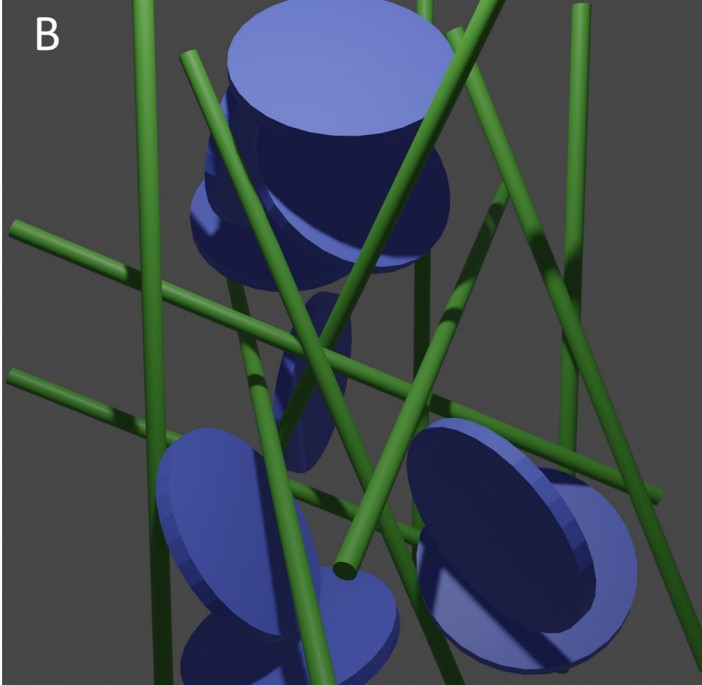

**Fig 7. Graphical illustration of the cellulose fibril and clay interactions.** Laponite (A) and montmorillonite (B) clay particles interact with the cellulose nanofibrils (green). Aggregates of the smaller laponite particles are more able to fit into and interact with the OCNF network than the much large aggregates of montmorillonite which leads to stronger interactions and more 'gel-like' properties.

caused a decrease in tan(δ) and an increase in viscosity (S34 and S35 Figs in the S1 File). This is attributed to electrostatic interactions between the negatively charged cellulose fibrils and the positively charged edges of the clay platelets. The negatively charged faces of the clay platelets could also contribute to gelation through repulsive stabilisation. It is therefore likely that addition of salts to these systems could affect the gel properties, as per addition of salt to OCNF on its own, which greatly increases viscosity [3]. This is relevant as many real-world applications will require the presence of free ions.

Addition of different clays i.e. Laponite or montmorillonite has a different effect on the rheological properties, with Laponite causing a more significant decrease in tan(δ) and increase in viscosity. This is probably due to the smaller platelet size of Laponite compared to montmorillonite, which not only allows them to connect the cellulose fibrils more easily, but also allows a greater surface area of (positive) edge charges that can interact with the negatively charged fibrils. The effect of particle shape on these interactions, for example the influence of aspect ratio on depletion forces [38], would be a valuable area for future study to better understand the interactions of differently shaped particles.

Increasing the concentration of the clay component resulted in a lower tan(δ) and higher viscosity. In all cases, the mixtures retained the shear-thinning behaviour of OCNF and therefore could be of great interest for industrial applications.

These results suggest environmentally-friendly, and renewable additives that could be combined with OCNF for rheological modification for use in personal and home care, as well as other industrial applications. While previous works have combined cellulose and clays for film and paper production [7–12,14–16], this is the first time that the authors know of that two different clays in combination with cellulose have been rigorously studied for hydrogel formation. This work can be used for more targeted development of specific home and personal care products, using these additives in place of the current, non-renewable materials.

## Supporting information

**S1 File. Detailed methods and characterisation of the components.** Frequency and amplitude sweep graphs, flow sweep curves, SAXS data and fitting parameters.
(PDF)

## Acknowledgments

The authors gratefully acknowledge the Material and Chemical Characterisation Facility (MC[2]) at University of Bath (https://doi.org/10.15125/mx6j-3r54) for technical support and assistance in this work.

This work benefited from the use of the SasView application, originally developed under NSF award DMR-0520547. SasView contains code developed with funding from the European Union's Horizon 2020 research and innovation programme under the SINE2020 project, grant agreement No 654000.

Data supporting this work are freely accessible in the Bath research data archive system at DOI: 10.15125/BATH-00791.

## Author Contributions

**Conceptualization:** Saffron J. Bryant, Janet L. Scott, Karen J. Edler.

**Data curation:** Marcelo A. da Silva.

**Funding acquisition:** Janet L. Scott, Karen J. Edler.

**Investigation:** Saffron J. Bryant, Vincenzo Calabrese, Marcelo A. da Silva, Kazi M. Zakir Hossain.

**Methodology:** Marcelo A. da Silva.

**Supervision:** Janet L. Scott, Karen J. Edler.

**Visualization:** Saffron J. Bryant.

**Writing – original draft:** Saffron J. Bryant.

**Writing – review & editing:** Saffron J. Bryant, Vincenzo Calabrese, Marcelo A. da Silva, Kazi M. Zakir Hossain, Janet L. Scott, Karen J. Edler.

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
