## [Decision Letter · Decision Letter 0]

2 Dec 2020

PONE-D-20-31757

Rheological modification of partially oxidised cellulose nanofibril gels with inorganic clays

PLOS ONE

Dear Dr. Edler,

Thank you for considering the PLOS ONE journal for submitting your work. It is an interesting contribution to the field. Despite the merits (experimental design and description) presented in manuscript, it needs further revision to improve the clarity as suggested by the reviewers. Therefore, we invite you to submit a revised version of the manuscript that addresses the points raised during the review process.

Looking forward to recieve your revised manuscript by Jan 16 2021 11:59PM. If you will need more time than this to complete your revisions, please reply to this message or contact the journal office at plosone@plos.org. Please include the following items when submitting your revised manuscript:

We look forward to receiving your revised manuscript.

Kind regards,

Pratheep K. Annamalai

Academic Editor

PLOS ONE

Additional Editor Comments:

This manuscript reports the rheological (gelation) behavior of oxidised cellulose nanofibre inluenced by two different types of clays which present platelet-like morphology. This is an important contribution to the field of nano-materials and applications, and suitable for the journal.

Before this can be considered for publication, it needs revision according to the reviewers' comments.

1. It would be good to add the detailed description in Supporting information on the preparation and characterisation of oxidised nanofibre (including source, experimental conditions, yield).

2. Were there any further characterizations performed on nanoclays such as CEC, pH in water or other that would be relevant for this study (apart from the platlet size). Ionic charges and strength on the surface might influence the interactions between nanofibers and clays. Correlations with such details in the discussion will enhance the clarity on structural information as indicated in reviewer comments.

3. Graphical representations of interactions (stable dispersion vs agglomeration) between nanofibres and two different clays (30 nm vs 300 nm) would be useful for the discussions.

Journal Requirements:

"SB, M.A.D.S., and K.M.Z.H thank EPRSC for funding this project (grant EP/N033310/1).

V.C. thanks the University of Bath for supporting his PhD.

"KJE (principal investigator) and JLS (co-investigator) received an award from the Engineering and Physical Sciences Research Council to fund this project, which employed SB, MADS and KMZH (grant no. EP/N033310/1) https://epsrc.ukri.org/. The funders had no role in study design, data collection and analysis, decision to publish, or preparation of the manuscript."

Reviewers' comments:

Reviewer's Responses to Questions

**Comments to the Author**

1. Is the manuscript technically sound, and do the data support the conclusions?

Reviewer #1: Partly

Reviewer #2: Yes

Reviewer #3: Yes

2. Has the statistical analysis been performed appropriately and rigorously? 

Reviewer #1: No

Reviewer #2: Yes

Reviewer #3: Yes

3. Have the authors made all data underlying the findings in their manuscript fully available?

Reviewer #1: No

Reviewer #2: Yes

Reviewer #3: Yes

4. Is the manuscript presented in an intelligible fashion and written in standard English?

Reviewer #1: No

Reviewer #2: Yes

Reviewer #3: Yes

5. Review Comments to the Author

Reviewer #1: The manuscript study the modification of the rheological properties of partially oxidised cellulose nanofibrils (OCNF) by the addition of Laponite or montmorillonite mixture. The main contribution of the article according to the authors is that it’s the first time that two different clays in combination with cellulose have been rigorously studied for hydrogel formation. However, there are the following comment.

1. The author claimed that “this work presents a mechanism for modifying rheological properties using renewable and environmentally-friendly nanocellulose and clays which could be used in home and personal care formulations”. However, the article did not carry out relevant experiments to demonstrate its mechanism, reproducibility, environmental friendliness and biocompatibility. It was suggested to characterize the composition, structure and other properties of composite.

2. The rheological properties of shear-thinning gels are affected by three factors: OCNF concentration, clay concentration and clay type. How did authors to design mentioned above three parameters to affect the rheological properties of gels.

3. In “Figure 1”, the author claimed that “it is clear that the montmorillonite suspension has a stronger frequency dependency than the Laponite suspension.” It needs further explanation and experimental evidence, or documentation.

4. In “Figure 2”, the author claimed that “Laponite contributed more significantly to the solid-like behaviour and that further addition of OCNF had little effect.” and “Laponite is a much smaller particle and therefore on a per-weight basis has a higher contribution to electrostatic interactions than OCNF fibrils”. It is difficult to get this conclusion directly from “Tan(δ) values of mixtures of Laponite and montmorillonite with OCNF”. The author's statement needs further experimental proof.

5. In “Figure 2”, the physical quantity on the abscissa lacks a logical relationship. It is recommended to redraw the diagram.

6. Please give more description of result, the equation or reference to calculate some index etc. to help readers to understand.

7. The abbreviation should be given its full name when it first appears.

Reviewer #2: The manuscript described the rheological modification by two different clays (Laponite and montmorillonite) in combination with cellulose nanofibers, and further the effect on hydrogel formation. The manuscript did a systematically study, and a detailed discussion of the rheological data. Before its consideration for publication, minor revisions are needed according to the following comments.

1. Please give a detailed description for the preparation of OCNF.

2. “the OCNF was freeze dried and then resuspended in deionised water to 2 wt%”, did the OCNF can be redispersed well without any sediments? As we know, the redispersing of dried cellulose nanofiber is still a big difficult.

3. Page 6, line 165-168, the author mentioned that “aging effect” could be observed for the combinations of Laponite and montmorillonite with OCNF in previous reported literature. If so, how about the combination of montmorillonite with OCNF? Are there any literatures, or any speculations from authors?

4. It is better to provide SEMs to observe the OCNF networks and the combination state with clays.

Reviewer #3: The authors present an interesting analysis of a mixed organic/inorganic gel comprising of oxidised cellulose nanofibres with two different types of clay particle. The system is probed with both rheometry and small-angle X-ray scattering, looking at both the structures that the fibre-clay gel forms and the flow properties of the gel/yielded fluid. The manuscript tells an interesting story that is suitable for PLOS One and will be more satisfying to the reader with some additional polishing as described below.

Comments

1. The origin of the interaction forces between the clay particles and the OCNFs needs to be described in the introduction so that the experimental data makes sense. It is established that all particles are negatively charged with the clays perhaps having a positive edge charge in some circumstances. The reader is left to assume that these are attractive gels (and not a repulsive gels) until some distance into the results before is it postulated that the positive edges of the clays is just attractive enough. Perhaps the authors could describe the gelation mechanisms a little better in the introduction.

2. Is there a role for shape in giving an attractive depletion potential, given the size and aspect ratio differences; this is a potentially (pun intended) extremely interesting aspect (pun intended) of this work given that the fibres have extremely large aspect ratios and are also so much better than the clay particles. Depletion potentials based on size/shape differences were demonstrated by Cinacchi et al. with both simulations and experiments.

3. The authors mention that the mesh size of the OCNFs might vary with salt but do not probe any salt effects. If an electrostatic interaction is needed for the gelation then salt is quite important. Can the authors point to suitable literature for this?

4. p13 L354 “As shown in Figures S24 and S25 ...”. The assertion that these figures justify a simple addition of the individual scattering components is lost on this reviewer; fits do not appear to be shown on those two figures at all. Indeed, fits do not appear to be included for a any of the SAXS data past Fig S23.

5. The SAXS data is comparatively featureless and yet the model used would have around 10 free parameters. The SAXS analysis could be more strongly presented if more details of the models were given in the SI, including which parameters were permitted to vary and which were held fixed based on extant data or previous experiments.

6. The volume fraction of the OCNFs must be quite large to have a 30nm pore size in the gel; what volume fraction does this correspond to? How does this volume fraction correspond to the dilute approximation that is used in the SAXS analysis where there is no structure factor included? While adding an additional 2-7 free parameters for a structure factor would not enhance the believability of the analysis, it seems unlikely that one is not needed.

7. The purpose of figures S24-S32 is unclear – these look to be interesting data but nothing seems to be done with them. There is minor reference to them at L361 but this probably needs to be expanded to an additional paragraph with systematic analysis of the data – if the scattering is genuinely additive as claimed then it should be relatively easy to extract some clustering or ordering parameters from these data.

8. This reviewer would love to see some additional effort made to link the structures and rheology that is contained in this manuscript. A little more explanation to help the reader understand how the specific structures measured by SAXS are driving the observed rheology would help the reader better understand the impact of this work. There’s a sentence that is doing this in the conclusions; and additional paragraph and perhaps a schematic for the end of the discussion would improve the manuscript.

Given this is journal does minimal typesetting, the authors are encouraged to:

9. p6 L137 & L139. Rewording this sentence so that it doesn’t start with “tan(δ)” is needed. (also in a few other places)

10. Throughout, ensure that the prime and double prime in G' and G'' are not curly quotation marks.

6. PLOS authors have the option to publish the peer review history of their article (what does this mean?). If published, this will include your full peer review and any attached files.

Reviewer #1: No

Reviewer #2: No

Reviewer #3: No

---

## [Author Response · Author response to Decision Letter 0]

13 Jan 2021

please see attached response to referees letter.

---

## [Decision Letter · Decision Letter 1]

8 Apr 2021

PONE-D-20-31757R1

Rheological modification of partially oxidised cellulose nanofibril gels with inorganic clays

PLOS ONE

Dear Dr. Karen J Edler,

Thank you for taking time to revise the manuscript and contribution to PLOS ONE. This work is highly important for the colloids and soft materials community and your contribution is highly appreciated. After careful consideration, we feel that while it has merit, there is still a room to improve clarity of presentation. Therefore, we invite you to submit a revised version of the manuscript that addresses the points raised during the review process.

We look forward to receiving your revised manuscript.

Kind regards,

Pratheep K. Annamalai

Academic Editor

PLOS ONE

Journal Requirements:

Additional Editor Comments (if provided):

Authors are appreciated for revising the manuscript carefully. Further minor revision is required for improving clarity and flow of presentation.

1. Fig. X is okay for single figure. For more than one figures, it is better to cite as 'Figures' or Fig. 1 and 2, instead of 'Figs'.

2. All the supporting information can be in one file (if possible). The citation of supporting information parts can be consistent such as "Figure SX or Table SX or Page SX in Supporting Information. Please insert page numbers for Supporting information file. Accordingly the figure captions can be revised.

Reviewers' comments:

Reviewer's Responses to Questions

**Comments to the Author**

1. If the authors have adequately addressed your comments raised in a previous round of review and you feel that this manuscript is now acceptable for publication, you may indicate that here to bypass the “Comments to the Author” section, enter your conflict of interest statement in the “Confidential to Editor” section, and submit your "Accept" recommendation.

Reviewer #1: (No Response)

Reviewer #3: All comments have been addressed

2. Is the manuscript technically sound, and do the data support the conclusions?

Reviewer #1: No

Reviewer #3: Yes

3. Has the statistical analysis been performed appropriately and rigorously? 

Reviewer #1: No

Reviewer #3: Yes

4. Have the authors made all data underlying the findings in their manuscript fully available?

Reviewer #1: Yes

Reviewer #3: Yes

5. Is the manuscript presented in an intelligible fashion and written in standard English?

Reviewer #1: Yes

Reviewer #3: Yes

6. Review Comments to the Author

Reviewer #1: 1. For comment 1, it still lack some experimental evidence or reference to confirm the mechanism for modifying rheological properties using clay, e.g. SEM or TEM to reveal the distribution of clay particles in OCNF gel, FTIR or Laser Raman or XPS to demonstrate the interactions between OCNF and clay platelets.

2. For comment 6, it’s means that it hard to understand Line 278 to 288. What is “The exponent (flow behaviour index)”,” The base number (flow consistency index)”. It is suggested to give more description of result, the equation or reference to calculate some index etc. to help readers to understand.

Reviewer #3: The authors have done an excellent job in improving this manuscript based on the comments and suggestions provided by each of the reviewers. They have certainly adequately address all of this reviewer's comments and even appeared to take the bad puns in their stride. This is a delightful piece of work and I look forward to see it in in PLOS ONE.

7. PLOS authors have the option to publish the peer review history of their article (what does this mean?). If published, this will include your full peer review and any attached files.

Reviewer #1: No

Reviewer #3: No

---

## [Author Response · Author response to Decision Letter 1]

21 Apr 2021

please see uploaded Response to reviewers file.

---

## [Decision Letter · Decision Letter 2]

20 May 2021

Rheological modification of partially oxidised cellulose nanofibril gels with inorganic clays

PONE-D-20-31757R2

Dear Dr. Edler,

Thank you for addressing the comments and revision. We’re pleased to inform you that your manuscript has been judged scientifically suitable for publication and will be formally accepted for publication once it meets all outstanding technical requirements.

Kind regards,

Pratheep K. Annamalai

Academic Editor

PLOS ONE

Additional Editor Comments (optional):

Thank you. Much appreciated for the time and revision.

Reviewers' comments:

Reviewer's Responses to Questions

**Comments to the Author**

1. If the authors have adequately addressed your comments raised in a previous round of review and you feel that this manuscript is now acceptable for publication, you may indicate that here to bypass the “Comments to the Author” section, enter your conflict of interest statement in the “Confidential to Editor” section, and submit your "Accept" recommendation.

Reviewer #1: All comments have been addressed

2. Is the manuscript technically sound, and do the data support the conclusions?

Reviewer #1: Yes

3. Has the statistical analysis been performed appropriately and rigorously? 

Reviewer #1: Yes

4. Have the authors made all data underlying the findings in their manuscript fully available?

Reviewer #1: Yes

5. Is the manuscript presented in an intelligible fashion and written in standard English?

Reviewer #1: Yes

6. Review Comments to the Author

Reviewer #1: (No Response)

7. PLOS authors have the option to publish the peer review history of their article (what does this mean?). If published, this will include your full peer review and any attached files.

Reviewer #1: No

---

## [Editor Report · Acceptance letter]

28 Jun 2021

PONE-D-20-31757R2 

Rheological modification of partially oxidised cellulose nanofibril gels with inorganic clays 

Dear Dr. Edler:

I'm pleased to inform you that your manuscript has been deemed suitable for publication in PLOS ONE. Congratulations! Your manuscript is now with our production department. 

Kind regards, 

on behalf of

Dr. Pratheep K. Annamalai 

Academic Editor

PLOS ONE